# Improving Hazardous Gas Detection Behavior with Palladium Decorated SnO_2_ Nanobelts Networks

**DOI:** 10.3390/s23104783

**Published:** 2023-05-16

**Authors:** Estácio P. de Araújo, Murilo P. Paiva, Lucas A. Moisés, Gabriel S. do Espírito Santo, Kate C. Blanco, Adenilson J. Chiquito, Cleber A. Amorim

**Affiliations:** 1NanOLaB, Departamento de Física, Universidade Federal de São Carlos—UFSCar, Rodovia Washington Luiz, Km 235 Monjolinho, CP 676, São Carlos 13565-905, SP, Brazil; estaciopaiva@gmail.com (E.P.d.A.);; 2Programa de Pós-Graduação em Engenharia Elétrica (Mestrado), Instituto de Ciência e Tecnologia-Câmpus de Sorocaba, Sorocaba 18087-180, SP, Brazil; 3School of Sciences and Engineering, São Paulo State University (Unesp), Av. Domingos da Costa Lopes, 780 Jardim Itaipu, Tupã 17602-496, SP, Brazil; gse.santo@unesp.br; 4São Carlos Institute of Physics, University of São Paulo, P.O. Box 369, São Carlos 13566-970, SP, Brazil

**Keywords:** nanobelts, tin oxide, gas sensor, carbon monoxide, carbon dioxide, CO, CO_2_

## Abstract

Transparent Conductive Oxides (TCOs) have been widely used as sensors for various hazardous gases. Among the most studied TCOs is SnO_2_, due to tin being an abundant material in nature, and therefore being accessible for moldable-like nanobelts. Sensors based on SnO_2_ nanobelts are generally quantified according to the interaction of the atmosphere with its surface, changing its conductance. The present study reports on the fabrication of a nanobelt-based SnO_2_ gas sensor, in which electrical contacts to nanobelts are self-assembled, and thus the sensors do not need any expensive and complicated fabrication processes. The nanobelts were grown using the vapor–solid–liquid (VLS) growth mechanism with gold as the catalytic site. The electrical contacts were defined using testing probes, thus the device is considered ready after the growth process. The sensorial characteristics of the devices were tested for the detection of CO and CO_2_ gases at temperatures from 25 to 75 °C, with and without palladium nanoparticle deposition in a wide concentration range of 40–1360 ppm. The results showed an improvement in the relative response, response time, and recovery, both with increasing temperature and with surface decoration using Pd nanoparticles. These features make this class of sensors important candidates for CO and CO_2_ detection for human health.

## 1. Introduction

Transparent Conductive Oxides (TCOs) have been studied for over a century, since Bädeker showed that a thin film of cadmium became transparent when oxidized [1]. TCOs have had a great technological appeal mainly due to their optoelectronic properties. Among the main conductive oxides, we can highlight tin oxide (SnO_2_), as it is an extremely versatile material, abundant, low cost, easy to handle, has low toxicity, and is crystalline and reproducible [2]. SnO_2_ has a tetragonal rutile structure belonging to the P42/mnm space group, good crystalline quality, lattice constants of a = b = 4.7399 Å and c = 3.1881 Å, high band gap energy (3.6 eV) and high exciton binding energy (130 meV) at room temperature [3,4,5,6,7]. SnO_2_-based gas sensors have been developed for years, mainly those whose conductance is modified due to interactions with the gas atmosphere [8,9,10,11,12].

In recent years, control of the electrical properties of biosensors, mainly of SnO_2_ nanobelts (NW), has attracted significant attention [12,13,14,15,16]. In particular, SnO_2_-based NW is particularly useful due to conductance tuning, possibly due to vacancy control [17]. Additionally, these materials have higher length-to-diameter and surface-to-volume ratios than bulk materials. SnO_2_ NW are one-dimensional and have attracted attention thanks to their unique properties and various special nanostructures. SnO_2_ is a strong candidate for future transistors, piezoelectric devices, and solar cells [18]. Furthermore, the high sensitivity to toxic and combustible gases can be highlighted, accelerating studies on SnO_2_ gas sensors in different geometries, such as thin films, heterojunctions, nanoparticles, and NW [19].

Nanobelts are synthesized by various techniques such as vapor-liquid-solid (VLS), hydrothermal, carbothermal reduction, and molecule-based chemical vapor deposition [20,21,22,23], and have been widely applied as gas sensors. Thus, the new generation of gas sensors will have to present characteristics such as an excellent cost-benefit ratio, high sensitivity and selectivity, and being stable [24,25,26]. The selectivity of detection of gas sensors based on transparent conducting oxide (TCOs) is still the subject of significant challenges for the development of sensory systems used in environmental monitoring, as well as industrial safety applications.

The oxide NW metal functionalization (Pd, Pt, rare earth, Ce, Cu, for instance), in addition to presenting economic advantages, is also a powerful way to improve the sensitivity and selectivity of the NW [19]. According to Chan et al., the improvement obtained through functionalization by metallic nanoparticles can be understood from different views, such as manipulation of the acid-based properties of the NW surface, change in donor density, catalytic promotion, and extension of the electron depletion region at the junction metal-semiconductor [27]. SnO_2_ NW have already been functionalized with different catalytic nanoparticles such as SiO_2_, palladium, copper, platinum, and lysine to improve the selective detection of other analytes, including acetone, sulfur dioxide, nitrogen dioxide, carbon dioxide, and hydrogen [28,29,30,31,32].

The detection process in SnO_2_ NW is based on the change in the material’s conductivity through the difference in the concentration of electrons caused by adsorption and desorption of the analyte (gas). When analyte comes into contact with the SnO_2_, its molecules react with the adsorbed oxygen anions, releasing/trapping electrons, and changing the sensor conductivity. Kolmakov et al. showed that in the presence of carbon monoxide, the pre-absorbed species such as O^−^_(s)_ and O^2−^_(s)_ reacts with the SnO_2_ surface, reducing the concentration of oxygen on the surface, and donating the few electrons to the volume of the material, resulting in an increase in conductivity that depends on the CO concentration [8]. In an atmosphere rich in nitrogen, commonly used as a carrier gas, oxygen molecules are excluded from forming oxygen species such as $O_{2}^{-}$, $O^{-}$ and $O^{2-}$ in the region close to the surface [33,34]. CO_2_ is a weak reducing gas, poorly ionized when it reacts with oxygen vacancies. The electron gain, due to the presence of CO_2_, creates a change in the carrier concentration, thus decreasing its general resistance, as observed in several works [6,18,29].

Gases such as CO_2_ and CO act directly and indirectly on climate change, in addition to having a great influence on the quality of life because they are harmful to human health. There are several myths about how these gases are obtained and the dangers they can cause. Both gases are present in the environment, have different compositions, and are generated through various chemical reactions. Additionally, CO_2_ and CO do not cause the same effects in the human body, although they are both toxic. CO can be produced through any incomplete combustion of fuel, usually associated with a low amount of oxygen. However, this gas is not formed naturally; it can be formed through boilers, combustion engines, heaters in general, fireplaces, etc. As can be seen, the generation of this gas is associated with human behavior, and as it is so dangerous, it can cause many problems for human beings, including death. According to the Occupational Safety and Health Administration (OSHA), concentrations above 200 ppm can cause headaches, fatigue, and nausea after 2–3 h of continuous exposure. Concentrations above 600 ppm can lead to death.

On the other hand, CO_2_ is also a toxic gas found in the atmosphere at a concentration of approximately 400 ppm, while the limited tolerance is around 3900 ppm. This gas can be generated, not only in the ways mentioned for CO, but also through fermentation processes. Due to the danger presented by these gases, their monitoring becomes essential, not only for safety but also for the quality of life, since even at low concentrations, there will be some adverse effects.

In the present work, NW made of pure SnO_2_ and decorated with palladium was synthesized by vapor-liquid-solid (VLS) methods, forming a network of nanobelts. Pd nanoparticles were deposited after NW growth by thermal evaporation. This process allowed us to build a simple and practical fabrication device that operates at temperatures close to ambient. The methodology presented in this work does not require the use of a clean room, requiring only nanobelt growth reactors, thermal evaporation, and a furnace for the formation of metallic nanoparticles. In this way, we believe we have a simple and easy-to-apply device. Both devices were tested at different temperatures, 50 and 75 °C. Their response was obtained for various concentrations of both CO and CO_2_. A comprehensive study of these gases’ detection mechanisms was also presented. Finally, the results showed a relatively low concentration of the studied gases compared to SnO_2_-based sensors in the literature, the vast majority of which work under high temperatures. Furthermore, our devices showed a rapid response and recovery after exposure to the analytes studied.

## 2. Materials and Methods

SnO_2_ NWs were grown using the method proposed by Wagner et al., Vapor-Liquid-Solid [20] and also described by Araújo et al. [35]. This method consists of depositing a thin gold metallic layer (15 Å) on an Si/SiO_2_ substrate, which will serve as catalyzed nanoclusters. These clusters are used as preferential sites for the adsorption of vapor phase components, leading to one-dimensional growth of the structures. A total of 0.1 g of tin powder (Aldrich, purity > 99.99%) was placed in an alumina crucible and placed in the center of a tubular reactor (Lindberg Blue M, Thermo Scientific, Walthan, MA, USA). The system is heated to 950 °C at a rate of 20 °C/min and held at this level for 50 min. The vapor generated by the precursor powder is carried by an Argon/Oxygen gas mixture (15/8 sccm- Mass Flow MKS 1149, Andover, MA, USA) throughout the synthesis time, and the pressure is kept constant at around 350 mbar; a schematic profile of the growth process is shown in Figure 1a. The material as grown is shown in Figure 1b. After growth, a palladium thin layer (~5 nm) was deposited by electron-beam onto the as-grown nanobelts, under a high vacuum (10^−6^–10^−7^ mbar). This evaporation presented a dark appearance in contrast to a lighter region shadowed by the sample holder used in the thermal coater system (Figure 1b). After the palladium evaporation (Figure 1c), these samples underwent heat treatment, 300 °C for 30 min, to produce the metallic nanoparticles (Figure 1d).

The material as-grown was analyzed according to scanning electron microscopy (SEM, Jeol 6510 Company, Peabody, MA, USA, operated at 20 kV), X-ray diffraction (XRD, Rigaku Company, Woodlands, TX, USA, D/max-2500, Cu-Kα radiation), and Raman Spectroscopy (Horiba-Jobin-Yvon, Ann Arbor, MI, USA, laser diode operating at 532 nm).

Once the material was grown, in which electrical contacts to nanobelts are self-assembled, the sensors do not need any expensive and complicated fabrication processes, such as photolithography processes. We emphasize that this is extremely important because it allows the development of devices quickly and at a lower cost when compared to those that require photolithography. Such types of devices were already used in previous works, as seen in [35,36,37,38]. These devices were built in a metal/NW/metal architecture, where the metallic electrodes were commercially ‘Test Probes Contact Pins’ type, connected directly to the gas detection chamber.

The target gases in this study were CO_2_ (40–800 ppm; 99.99% purity) and CO (70–1300 ppm; 99.99% purity). The different desired concentrations were obtained by purging the gas in an airtight chamber of approximately 50 mL with a known volume of nitrogen (99.98% with 10 ppm H_2_O) gas determined by a mass flow controller. The concentration obtained is a ratio between the nitrogen fluxes (background gas) and the target gas (CO and CO_2_), controlled by mass flow controllers and solenoid valves, in addition to the respective molar masses of these gases. That is why the concentration values were so different from each other. The final gas flow was maintained at a rate of approximately 50 sccm. Teflon O-rings were used in the metal joints of the gas chamber to seal it against gas leaks. Sensor experiments were conducted under nitrogen flow with different conditions: i. pure SnO_2_, different concentrations of target gases, temperatures of 50 and 75 °C; ii. SnO_2_ with palladium nanoparticles, different concentrations of target gases, and temperatures of 50 and 75 °C.

Figure 1e shows the characterization configuration of the target gas sensors. All measurements were made under nitrogen gas as a background, to which the target gases were mixed. The measured concentrations were obtained using a mass flow controller of one cubic centimeter per minute (sccm). The detection chamber has a volume of approximately 25 cm^3^; coupled to it, there is a heater where the working temperature can be controlled from room temperature to 300 °C. Attached to the detection chamber, two connection pads make electrical contact with the NW network. This process demonstrates the practicality and speed of building and testing gas sensors. Current-time characterization was performed by applying an electrical voltage of 5.0 V and monitoring the electrical current using a Keithley 6517-B electrometer (Keithley, Cleveland, OH, USA). Finally, a sketch model of the applied voltage is shown at *t*_0_ = 0; after a specific time, the target gas is inserted into the camera, and the sensor responds as shown in the red curve. The figure refers to the result for an oxidizing gas (CO). After *t*_1_ > 0, the target gas is switched off, and the sensor behavior is restored. A nitrogen constant flow is maintained in the chamber where the gas is continuously pushed to the gas outlet.

The sensor relative response reflects the concentration of the target gases, and this can be monitored by recording the changes in the conductivity of the SnO_2_ nanobelt network. The sensors relative response was calculated as follows:(1)$$S=∆R/R_{0}=\frac{R_{0}–R_{E}}{R_{0}}\times 100,$$
where the *E*-index refers to resistance and current values under gas exposure, while the 0-index refers to the reference resistance/current values (baseline) which were obtained when the sensor is exposed to nitrogen flow only. Equation (1) has two exciting aspects depending on the exposed gas. When exposed to CO_2_, a reducing gas (it will transfer electrons to the nanobelts’ system), it will increase the sample current (S > 0). On the other hand, when exposed to an oxidizing gas such as CO, electrons will be withdrawn from the nanobelts’ system, decreasing the sensor current (S < 0). In this work, we define the response time where the sensor reaches 90% of the maximum change after exposure to the gas samples. Thus, recovery time was defined as the time required to recover 10% of the initial baseline value after exposure to the target gas [39].

## 3. Results

Figure 2a shows an SEM image of the samples as grown, evidence of the nanobelt network forming the device. At the bottom (Figure 2b) is a nanobelt with a catalytic Au nanoparticle at the tip, confirming the VLS growth mechanism. The SAED pattern (Figure 2c) confirms that the nanobelt is a single crystal with a tetragonal unit cell. The crystalline phase of the synthesized samples was analyzed by X-ray diffraction. Figure 2d shows the spectrum obtained at room temperature for pure SnO_2_ nanobelts. The results indicate that the samples presented a tetragonal structure of the rutile type (JCPDS: 41–1445) belonging to the P42/mnm space group [3,5]; the nanobelts showed an excellent crystalline quality. The composition of the as-grown material can be shown through EDX measurements. These results show that the nanobelts are composed of Sn and O (Figure 2e). Finally, Figure 2f shows the Raman spectrum of the SnO_2_ nanobelts. We can observe that the results showed the bands centered at 474, 632, and 775 cm^−1^, corresponding to three active Raman vibration modes (E_g_, A_1g_, and B_2g_), respectively [36,37]. The results showed a typical characteristic of the rutile phase of SnO_2_ nanobelts in agreement with the XRD measurements.

Figure 3 presents FEG-SEM images comparing pure SnO_2_ nanobelts and those decorated with palladium nanoparticles (5 and 50 nm). Additionally, the EDX spectrum for each one of them and their respective quantifications are presented. Figure 3a shows a micrograph for SnO_2_ nanobelts as grown; it is possible to observe a smooth aspect of the nanobelts. The EDX spectrum is shown just below, the silicon peak is due to a substrate response. The other peaks refer to oxygen and tin only, as shown in the table next to the figure. The quantification shows that only oxygen and tin are present, the quantification of palladium, in this case, was forced only for comparison with the other materials. On the other hand, Figure 3b shows the micrograph for SnO_2_ nanobelts with a 5 nm Pd evaporation, followed by a thermal treatment, as described before. In this case, we have already observed a different aspect on the surface of the nanobelts. Several dots are formed on the surface, showing the presence of Pd nanoparticles. This aspect is confirmed through the appearance of the peak referring to Pd in the EDX spectrum at approximately 2.8 keV and the percentage of approximately 1% in atomic mass in the composition of these nanobelts. Finally, Figure 3c presents the micrograph for a SnO_2_ nanobelt with a 50 nm layer of Pd. It is possible to observe in the EDX spectrum a very accentuated peak of palladium, corroborated by an amount of approximately 18% atomic mass of this material in the formation of SnO_2_ nanobelts.

Figure 4a shows the relative response curve obtained from Equation (1) at room temperature for the CO sample. The concentrations studied were extremely high, above 1600 ppm. The system was tested for these values because they were the only ones that offered the sensor response at room temperature. Figure 4b shows the curves obtained when the target gas is carbon dioxide in concentrations between 960–3000 ppm. The concentration obtained is a ratio between the nitrogen fluxes (background gas) and the target gas (CO and CO_2_), in addition to the respective molar masses of these gases. That is why the concentration values were so different from each other.

There are two essential aspects of Figure 4a,b: *i.* the sensor responses of both curves are practically the same, although with a different shape; *ii.* And the relative response obtained when the target gas was CO was expected to have a negative relative response in Equation (1) because it is an oxidizing gas that steals electrons from the sensor surface (this process is described below). To obtain any signal beyond the background, the gas flow inside the detection chamber needed to be above 100 sccm. Indeed, when the target gas is CO_2_, there is some charge transfer to the SnO_2_ nanobelt film. However, this transfer is irrelevant since the response obtained for a concentration of 3000 ppm is lower than for the other concentrations as seen in Figure 4b. Additionally, when the target gas was CO, the relative response was expected to be negative in Equation (1), as the response current is smaller than the background current, and this was not observed, as shown in Figure 4b. We believe that the response observed in both cases is due to turbulent flow (high sccm) inside the chamber, which is more likely related to a temperature variation or surface cleaning than to the presence of gas.

The results presented in Figure 4a,b led us to two questions: *i*. at what temperature can we have a correct answer for the gas sensors; *ii.* and under which conditions we will have the lowest sensor operating temperature. Figure 5 presents the relative response curves obtained for the target gases CO and CO_2_, for temperatures of 50 and 75 °C. All measurements were performed under a constant flow of nitrogen for approximately 120 s. After this period, the valve with the target gas is opened for approximately 30 s and then closed. Two cycles were obtained, the second cycle being more stable than the first.

The CO sensor response at 50 °C in Figure 5a shows a response with an increasing concentration of the target gas. It is noteworthy that the negative sign of the relative response is kept only to indicate the typical response of an oxidizing gas. As we increase the concentration, the sensor response tends to increase, except for values up to 1300 ppm. Such behavior can be attributed to sensor saturation, although in this case, it was expected that the observed relative response value would be close to 50%. Another explanation may be related to the one presented above. For these CO concentrations, the flux inside the detection chamber is relatively high to the point of lowering the working temperature and having a cleaning effect on the nanobelts’ network surface. Figure 5b shows the CO gas sensor response for a temperature of 75 °C in a concentration range between 136–1360 ppm. There is an evident improvement in the sensor response with increasing concentrations of the target gas; however, for high concentrations, there is still a saturation region. The entire process of detecting the SnO_2_ nanobelt film for carbon monoxide will be discussed later.

Figure 5c,d presents the results for the CO_2_ target gas (80–800 ppm) at 50 and 75 °C, respectively. We can see that the response practically doubled with the increase in temperature; in addition, there was an improvement in the resolution of the sensor, especially when we look at the second cycle. For a temperature of 50 °C, there is a response for the target gas; however, there is no direct relationship between the response and the increase in concentration. This effect is overcome when the temperature is 75 °C, when the definition of the response becomes evident. This process will be discussed later.

The results presented in Figure 4 and Figure 5 showed that although the SnO_2_ nanobelt network could detect the presence of the analyte under study, it was impossible to have a direct relationship and resolution, as found in the literature [6,8,40]. Increasing the working temperature (room temperature, 50 °C, and then 75 °C) improved these aspects, but the device still showed a low resolution. Similar to other authors who studied SnO_2_ as an active layer for detecting gases such as CO and CO_2_, we decorated the nanobelts’ films with palladium nanoparticles [29,41,42,43].

To improve the properties of SnO_2_ nanobelts-based sensors at a temperature as close as possible to room temperature, we decorated the nanobelts’ films with palladium nanoparticles, as described in Section 2. Responses were measured for both CO and CO_2_ over a wide concentration range. Figure 6 shows the result for these samples for 50 °C (other temperatures were not tested because we are only interested in the working temperature closest to the ambient). Figure 6a shows the relative response curve at different concentrations. We observed that there is a better definition in the sensor response, that is, there is no more overlap between the response curves when compared to the results presented in Figure 5a,b, with the maximum response being more than doubled. Figure 5b shows the detection spectrum of the SnO_2_ nanobelt film sensor, taken by increasing the concentration at each cycle. There is a low background response when the concentration is below 210 ppm (third “peak” from left to right in Figure 6). From this value onwards, the response becomes more prominent and tends to increase with increasing CO concentration, and there is a saturation tendency for concentrations above 1150 ppm (17th peak).

Similarly, we tested the same device for the CO_2_ target gas; Figure 6c shows that the relative response increased about sixfold when compared to that shown in Figure 5c, the device without the palladium nanoparticles. The increase was 2.5 times higher when compared to the sample without the nanoparticles at 75 °C. Additionally, we observed an improvement in the sensor resolution, with a more straightforward response for different concentrations used. Figure 6d shows the response obtained for the CO_2_ target gas at a concentration of 40–800 ppm with steps of 40 pm. For concentrations below 120 ppm, the response is very low and close to the background values. However, for values above this, the response becomes clearer, tending to saturation for values above 680 ppm. In the next section, we will discuss in more detail why the sensor is sensitive to these two gases and why the presence of nanoparticles improves the relative response and resolution of the sensor.

Some authors have studied the behavior of CO and CO_2_ sensors at room temperature. Naama et al. studied the behavior of silicon nanowires in the detection of CO_2_, however comparison with this work is difficult because they are not SnO_2_ nanobelts and because the concentration is presented in pressure units [44]. Nandan Singh et al. presented results at room temperature for the detection of CO gas for samples of Zn:In_2_O_3_ field effect transistors, although the detection limits observed by the authors are lower than those observed in this work; the authors used a device more complex and requires more production steps [45]. We emphasize that the device used in this work is of the label-free type and that the steps to be considered for its use are only growth of the nanowires, evaporation of the nanoparticles and thermal treatment for the formation of the nanoparticles.

Some parameters are extremely important to characterize a sensor, such as relative response (here defined as the change in the input parameter required to produce a standardized output change) and the response and recovery time of the sensor. These parameters can place the device as a strong candidate for future applications. Figure 7 compiles all this information regarding the characterization parameters of the CO and CO_2_ sensor based on the SnO_2_ nanobelts film.

Figure 7 shows the results for response and recovery time, relative response vs. concentration obtained for the curves shown in Figure 5 and Figure 6. The solid lines represent the response time, and the dashed lines the recovery time. We have two comparisons to be carried out between the two temperatures studied and between the sensor with and without palladium nanoparticles. The response time obtained was between 160–20 s. In all cases, the response time tends to decrease with the increasing gas concentration. For a temperature of 50 °C (solid line-circle), there is an increase in response time up to approximately 350 ppm. From this point on, there is a considerable drop, tending to a constant response time for concentrations above 800 ppm. Similar behavior was observed for sensor response at 75 °C (solid line-square); however, a shorter response time was observed for this case, and from 200 ppm, no significant change was recorded. When we compare the two temperatures studied, we can observe that the response time is influenced by them for concentrations below 800 ppm; for values above that, both converge.

On the other hand, the presence of nanoparticles makes the response time of the sensors shorter when compared to the device without nanoparticles for the same temperature. Here we define the response time as the one where the sample recovers 90% of its initial value. Observing the graphs shown in Figure 5a–d, it is possible to estimate that the regeneration time is around twice the response time.

For values above 800 ppm, the value is close to the other cases, but getting slightly higher. The influence of temperature and nanoparticles decorating the nanobelts film is best observed in the recovery time of the sensors (red dash lines). The best recovery times were observed for the nanobelts film with palladium nanoparticles, with a practically constant value with increasing concentration (~120 s). When comparing sensor operation at 50 and 75 °C, the latter showed a slightly longer recovery time when comparing the exact gas concentration. The effect of temperature and the sensor with nanoparticles was also evident in the response time of the sensor when exposed to CO_2_ gas (see Figure 7b). Above 150 ppm, the response time was practically constant at 20 s, half the response time compared to CO gas. Additionally, the recovery time was not influenced by the presence of nanoparticles, nor by the increase in temperature. Here, we define the response time when the sensor reaches 90% of the maximum observed response. In cases involving toxic gases, such as the ones we are studying, often the response time must be the one in which the sensor reaches 50% of the maximum value. In this context, the times observed in this work presented a fast response, being very useful in alarm situations where a complete response is unnecessary.

Figure 7c,d presents the results for relative response vs. concentration. This information shows us how sensitive the sensor can be. Aside from the response for the CO sensor at 50 °C, all other measurements showed an improvement in relative response with the increasing gas concentration. A linear response was observed for concentrations above 250 ppm for CO gas and above 150 ppm for CO_2_. The relationship between relative response and concentration follows a linear fit (S=a+b·C), where a is the intercept of the y-axis; here, it does not have much meaning. b is the sensitivity growth rate with concentration (%/ppm), and C is the target gas concentration. Table 1 shows the values obtained for this adjustment under different experimental conditions. The presence of nanoparticles improved the response rate for both gases, with an increase of 26 times for CO and approximately 4 times for CO_2_, respectively.

## 4. Discussions

All curves based on the change in sensor resistance when exposed to target gas can be divided into three phases: stabilization → adsorption/response → recovery. The surface of the nanobelts plays a crucial role in the response of the gas sensor based on metal oxides. As already discussed in another work [33,34,35], tin oxide is one of the most studied n-type semiconductor TCOs for gas sensors. Its detection mechanism is explained based on the change in electrical conductivity, which occurs through the chemical interaction of gas molecules with the oxide surface [46]. In the presence of the background gas, lesser electrons get enough energy to jump into the conduction band and travel across the junction while holes are left behind. There will be an accumulation of charges at the interface, forming an intermediate spatial charge layer. Thus, the gas sensor will not change resistance when its surface is saturated with nitrogen. As a result, a depletion region is formed. Under the atmosphere of the gas, the acceptor or donor electrons are adsorbed on the surface of the metal oxide, resulting in a change in the material’s conductivity. The conductivity of the TCO’s gas sensor depends on the charge transfer mechanisms between the adsorbed gaseous species, as well as the surface reaction with the gas. Figure 8 shows the charge transfer on the sensor surface, as well as a diagram representation of the proposed bifunctional detection mechanisms, both for the effect of oxidizing gases and for reducing gases [47].

In our study, we have a SnO_2_ nanobelt network which can be described mainly on the surface of the nano and at the junction between the nanobelts that make up the network. Figure 8a represents this band configuration for this SnO_2_ nanobelts network, where the contact point between two nanobelts has the same behavior as a grain boundary structure. These interconnected NW form larger aggregates connected by these junctions. As previously announced, oxygen molecules play a crucial role in the SnO_2_ nanobelt properties. Thus, the detection mechanism is governed by oxygen vacancies on the surface of SnO_2_ [48,49,50]. Figure 8b shows the condition of the nanobelt network in the presence of background gas. The red spheres represent the oxygen molecules. The electrons that are available in the conduction band can be captured by the oxygen species. Due to these electrons, a depletion region ($d_{1}$) and a potential barrier ($\phi_{1}$) appear on the surface, preventing the movement of electrons between the NW junctions. When a reducing agent, such as CO_2_, is introduced into the system, it interacts with the oxygen species, causing a discharge of electrons from the surface of the NW, previously linked to the oxygen species. Therefore, the electrons are now free to return to the conduction band. Due to this charge transfer phenomenon, the depletion region ($d_{2}$) tends to reduce (compared to the situation without the presence of the target gas) and the electron density increases. In this condition, the potential barrier is represented by ($\phi_{2}$), and it is noted that it almost disappears in the presence of gas, and the device performance is improved. The contribution of these surface states must be considered on all NW surfaces. This leads to non-negligible band bending near the NW surface, resulting in an effective energy barrier [51].

The performance of the devices will depend on the electronic activity on the semiconductor surface and on the amount of the chemisorbed oxygen species (O^−^ and O^2−^) of the target gases. Wetchakun et al. and Yuliarto et al. showed how the CO_2_ detection mechanism in SnO_2_ can be. When SnO_2_ is used to detect CO_2_, a reaction occurs on the surface of SnO_2_ between CO_2_ gas and the oxygen species [6,52]. This process can be expressed as:(2)$$\left\{\begin{array}{c} CO_{2}\left(gas\right)+e^{-}\to CO^{2-}\left(ads\right)\\ CO^{2-}\left(ads\right)+O^{-}\left(ads\right)+{2e}^{-}\to CO\left(gas\right)+{2O}^{2-}\left(ads\right) \end{array}\right.$$

The gain of electrons from the gas causes a change in the carrier concentration, increasing the sensor’s conductivity. This behavior agrees with what we observed (inferior insert in Figure 8(b.1)) and is also observed in other works [29,33,53].

On the other hand, when an oxidizing gas (Figure 8c) comes into contact with the nanobelt network, such as CO, the oxygen species induces an increase in the depletion region (d3) and an increase in the potential barrier at the junction. In this case, the CO gas interacts with $O_{ads}^{-}$ and tends to release electrons in the conduction band, thus decreasing the conductance of SnO_2_ as seen in Figure 8(c.1), and as observed by different authors [54,55,56]. The equations below show the chemical reactions in this situation [47].
(3)$$CO+O_{2,ads}^{-}⇌CO_{2,gas}+e^{-}$$
(4)$$CO+{2O}_{ads}^{-}⇌CO_{3}^{2}⇌CO_{2,gas}+{O_{ads}^{-}+e}^{-}⇌CO_{2,gas}+\frac{1}{2}O_{2}+{2e}^{-}$$

The Pd nanoparticles act as catalysts accelerating the dissociation of oxygen molecules, increasing the flow of oxygen ions adsorbed on the NW surface. The more oxygen ions adsorbed on the surface of the SnO_2_, the more detection sites there will be, leading to an increased relative response. When palladium nanoparticles are present in SnO_2_, oxygen ions are more easily adsorbed onto the metal ions forming O^2−^. Consequently, the number of electrons decreases, the space charge region expands, and the initial resistance decreases. On the other hand, more electrons can be recovered when exposed to the target gases, so the response increases. At the same time, since it is easier to adsorb oxygen ions even at a lower temperature, we can lower the working temperature of the device.

Additionally, we can discuss the device response and recovery times (Figure 7a,b). With the different adsorbent sites and defects, we can expect different mechanisms of oxygen desorption, which contributes to a longer response time until the system reaches stability. Furthermore, the operating condition of the devices, close to ambient temperature, also plays an essential role in longer response times [48]. Araujo et al. demonstrated the influence of the applied electric field on the response and recovery time of SnO_2_-based sensors. In this case, there is a higher density of electrons that becomes available after cutting off the flow of the target gas due to a thicker depletion layer. When the gas flow is interrupted, there is a condition that facilitates a greater oxygen uptake [35]. Wen-Chieh Wang et al. showed that the response time in a hazardous sensor is related to the interactions of the gas with the surface. These interactions depend more on the effect of temperature than on different concentrations of the target gas (Figure 7a,b) [57].

SnO_2_-based sensors are widely studied because tin is an abundant, versatile, and stable chemical element. For example, Q. Wan et al. used antimony doped SnO_2_ to detect ethanol. In this work, the authors presented an operating temperature of 300 °C, detecting the target gas in the range of 10–200 ppm [10]. As for detecting the gases in this work, we can compare the response and recovery times observed by Tangivala et. al. with ours [30]. In that work, they found times of 20 and 30 s, respectively, while our response time was in the same order of magnitude and the recovery time was 6 times longer. The difference is that while our device was pure or decorated with nanoparticles, theirs were doped with copper and the operating temperature studied was between 100–300 °C. Similarly, Dandan et al. observed similar times at 240 °C and shorter times, 5 and 10 for response and recovery times, respectively, at 300 °C [33]. Using LaOCl-doped SnO_2_ nanofibers, Ya Xiong et al., found times similar to ours and lower than those observed by Dandan et. al. at 300 °C [53]. Other works that found times of the same order or greater than ours can also be highlighted [41,43]. However, they all use processes that require higher working temperatures than ours or more robust device preparation processes. A highlight can be given to the work of Ping et al.; the authors used field-effect transistors with an active layer of SnO_2_ decorated with palladium nanoparticles to detect respiratory gases (CO_2_ and O_2_). The detection range is between 0–2500 ppm at room temperature. Although these results may seem better than those presented here, the device requires a more elaborate manufacturing process, as it uses a gated terminal [54].

Our device has shown excellent performance as both a CO sensor and a CO_2_ sensor, both at the concentrations and temperatures studied. For example, Erin Stuckert et al. studied the effect of Ar/O_2_ and H_2_O plasma treatment on the performance of SnO_2_ nanobelts as a sensor [58]. The sensor’s relative response at different temperatures in a range of 25–300 °C was around 200% for a concentration of 100 ppm of CO at 300 °C. Nguyen Van Hieu et al., using a network of pure SnO_2_ wires and functionalized with LaO_3_, achieved a 3.3-fold response for a CO concentration of 100 ppm at 400 °C [59]. A similar result was obtained by the same authors in another work [60]. Additionally, Sung Hwang et al. obtained a response of approximately 140% in 100 ppm of CO for a network of SnO_2_ nanobelts at 450 °C [61]. All these results, obtained with a device similar to the one presented in this work, showed good sensory responses. However, they were obtained at temperatures above 300 °C, much higher than that used here. Similar results can be found for nanobelt-based sensors for CO detection, such as those presented by Brunet et. al. the authors compared the performance of SnO_2_ thin films with that of SnO_2_ nanobelts. They found a response around 0.5% at 350 °C to a CO_2_ concentration of 260 ppm [62]. Following the same idea, Trung et. al. compared the performance of a network of pure SnO_2_ nanobelts, and functionalized it with LaOCl. Although the functionalization improved the performance of the devices, it did not decrease the operating temperature, which was in the range of 300–450 °C [63]. These comparisons highlight some important aspects of the device presented in this manuscript, such as the wide detection range (136–1360 ppm for CO and 80–800 ppm for CO_2_), in addition to an operating temperature very close to ambient temperature (50 °C).

The processes described above allowed us to study and understand the detection mechanisms of our device, showing that oxygen species play a crucial role in this process. Furthermore, our devices showed exciting characteristics compared to others found in the literature, such as wide detection range, good response time, presence of an active layer without doping, and low operating temperature. All these characteristics are even more remarkable when we compare the manufacturing process of our sensor, which is practical and fast, as it does not need processes that involve a clean room environment, chip construction for interrogation, and/or doping processes that can be laborious and non-uniform.

## 5. Conclusions

In summary, the results presented showed a device based on a network of SnO_2_ nanobelts—grown by the VLS method—that is promising for detecting hazardous gases. This is due to the ability to build the device quickly and without the need for photolithography processes, eliminating the need to use a clean room. We could show a wide detection range for both CO and CO_2_, at values below concentrations that could present a health risk. Additionally, the responses for both gases studied were more significant than 50% for CO and 20% for CO_2_. These results were improved by increasing the temperature from 50 to 75 °C by almost twice. Decorating the devices with palladium nanoparticles improved not only the sensor’s response but also its resolution. In this situation, it was possible to observe that the responses reached 250% for CO and 120% for CO_2_, respectively. Furthermore, we found that both the response time and the recovery time are low enough to be applied as detectors for the presence of these hazardous gases. Finally, we discuss the models applied to explain the detection mechanisms of both target gases.

## Figures and Tables

**Figure 1 sensors-23-04783-f001:**
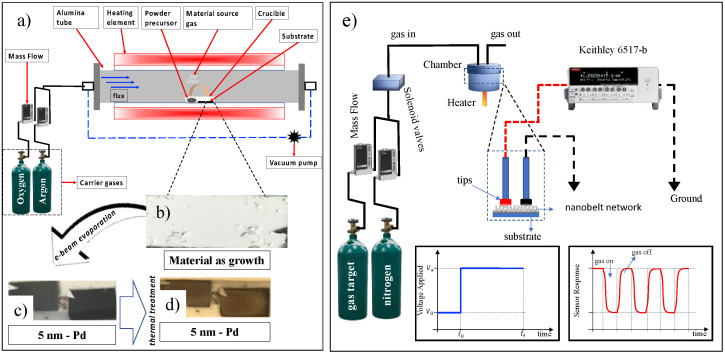
Schematic representation of the nanobelts’ growth configuration, decoration with nanoparticles, and structural characterization: (**a**) profile of the tubular furnace reactor with the main items for the growth of nanobelts, the symbol * stands for the point where the vacuum pump is inserted; the material as grown on Si/SiO_2_ substrate is shown in (**b**); in (**c**), some as-grown samples went through a process of deposition of palladium nanoparticles (5 nm) and it is possible to observe a white region due to the sample holder inside the thermal coater system; in panel (**d**), the substrates with the nanobelts after the heat treatment are shown, and (**e**) is the schematic representing the setup configuration used for gas detection, including gas cylinders, mass flow controllers, solenoid valves, detection chamber, test leads, the Keithley 6517-B electrometer (Keithley, Cleveland, OH, USA), and a representative response curve sensor (oxidizing gas).

**Figure 2 sensors-23-04783-f002:**
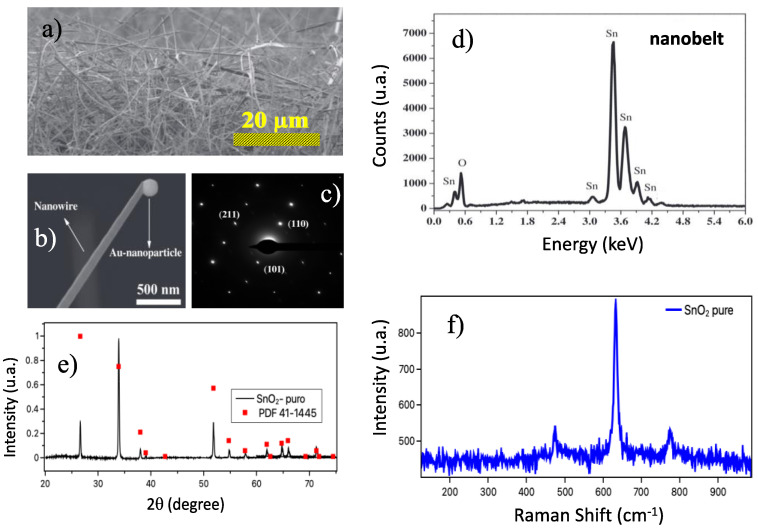
Samples shown in panel were characterized by morphological and structural analysis: (**a**) in panel is depicted an SEM image showing a network of as-grown nanobelts, (**b**) highlights a single nanobelt of SnO_2_ showing the Au nanoparticles, typical of VLS growth (immediately below figure); (**c**) HRTEM SnO_2_ Rutile phase (Laue Pattern); (**d**) the EDX spectrum of the as-grown samples is shown; and in panels (**e**,**f**), the XRD pattern of the as-synthesized SnO_2_ nanobelts and room-temperature Raman spectrum of SnO_2_ nanobelts are shown.

**Figure 3 sensors-23-04783-f003:**
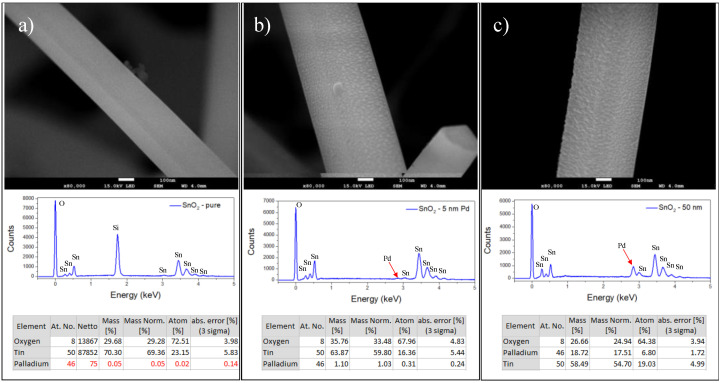
For each vertical sequence, (**a**) a typical FEG-SEM image of the SnO_2_-nanobelt sample as grown is presented, followed by the respective EDX spectrum, in which it is possible to observe the peaks referring to tin and to oxygen, and finally a table showing the compositions of the analyzed material; in (**b**) the same analyses are performed for a SnO_2_-nanobelt sample with a 5 nm palladium layer, while in (**c**) the results are obtained for a SnO_2_-nanobelt sample with a 50 nm palladium layer.

**Figure 4 sensors-23-04783-f004:**
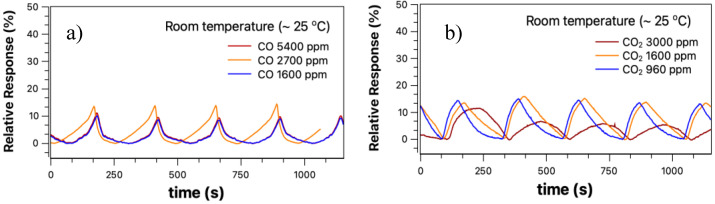
(**a**) sensor response for the CO target gas at different concentrations, and (**b**) the relative response for CO_2_ for room temperature (25 °C).

**Figure 5 sensors-23-04783-f005:**
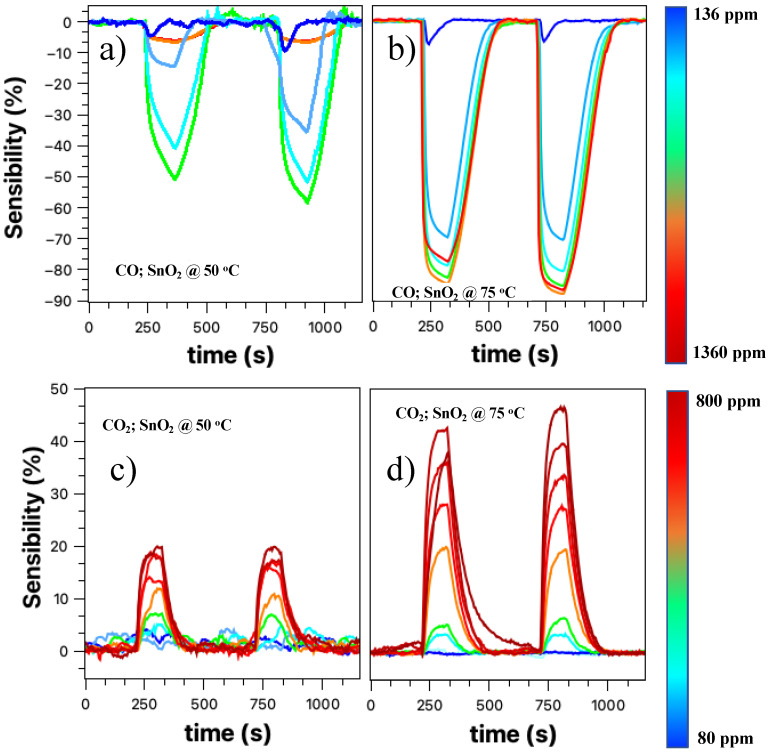
Panels (**a**,**b**) show sensor relative responses for CO gas at temperatures 50 and 75 °C, respectively. In panels (**c**,**d**), the sensor relative responses for CO_2_ gas are shown at 50 and 75 °C, respectively.

**Figure 6 sensors-23-04783-f006:**
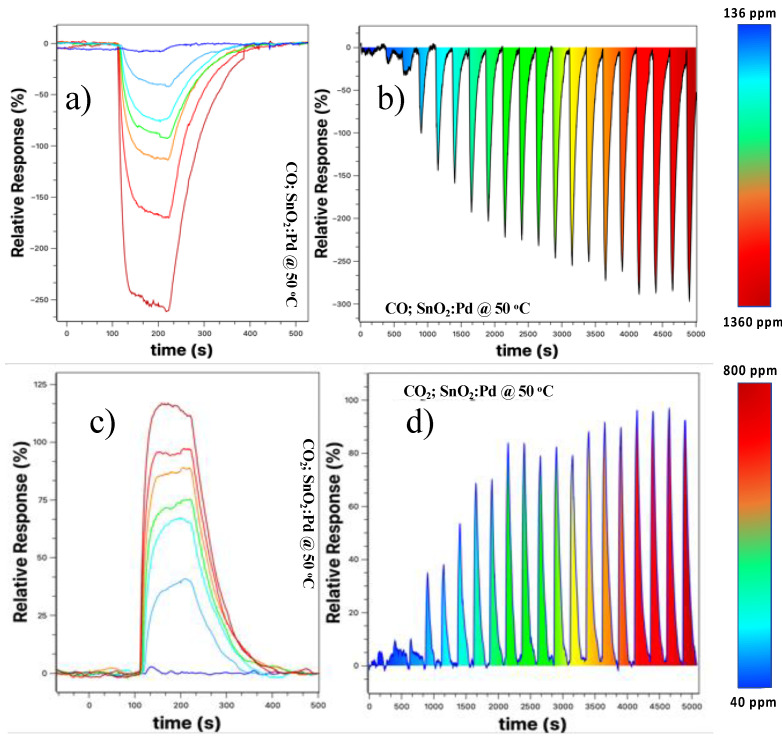
(**a**) Sensor response for different CO concentrations within 136–1360 ppm; (**b**) Similar measure to the previous one. However, the concentration was gradually increased after each cycle. In (**c**) we have the relative response for different CO_2_ concentrations between 40–800 ppm, and in (**d**) a gradual measure of the relative response for the described range of concentrations. All are under the same applied voltage value of +5 V.

**Figure 7 sensors-23-04783-f007:**
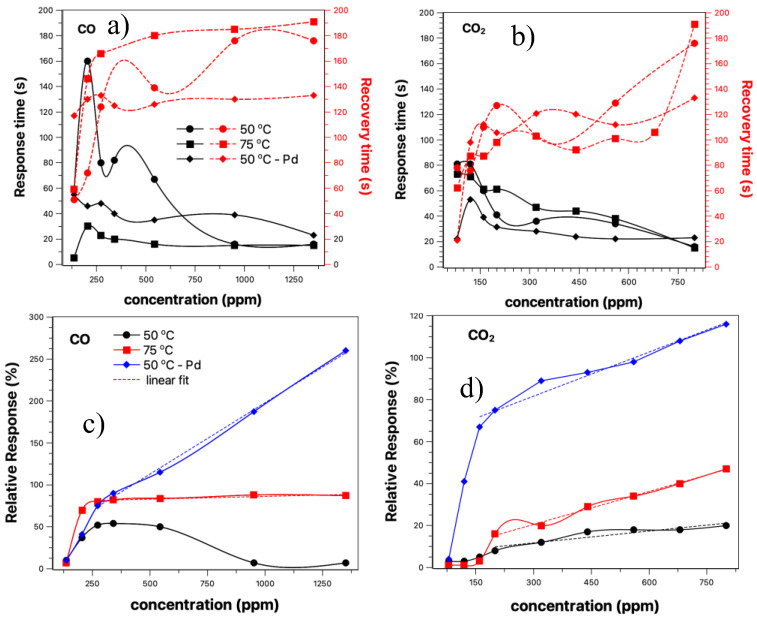
Response time and recovery time for CO and CO_2_ gas sensor, (**a**,**b**), respectively. In (**c**,**d**) we have the relative response vs concentration graphs for CO and CO_2_, respectively. All the results presented were taken for 50 °C with and without palladium nanoparticles and 75 °C. Legend: (■) 50 °C; (●) 75 °C; (◆) 50 °C–Pd nanoparticles.

**Figure 8 sensors-23-04783-f008:**
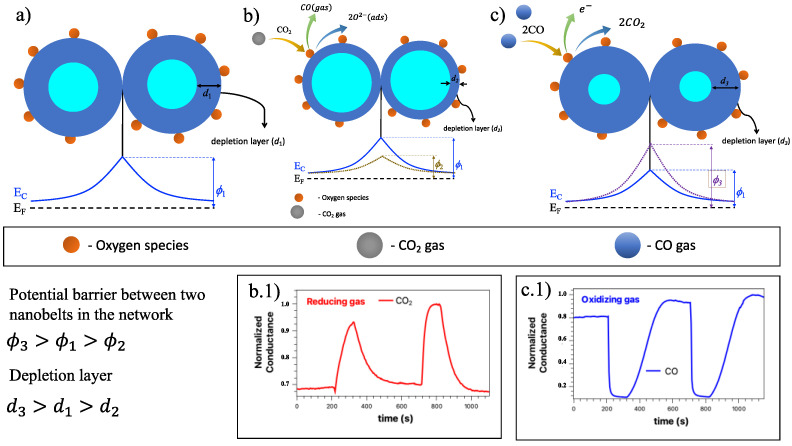
Schematic illustration of the chemical reactions on the surface of the nanobelts and the respective band diagram: panel (**a**), the network of nanobelts with oxygen molecules adsorbed on the surface of the nanobelts; panel (**b**), the nanobelts network surface under CO_2_ gas, and (**b.1**) conductance vs. times under this circumstance; in (**c**), the nanobelts network surface under CO gas and (**c.1**), conductance vs. times under this circumstance.

**Table 1 sensors-23-04783-t001:** Table referring to the parameters for adjusting the curves shown in Figure 7c,d.

	CO			CO_2_		
Temperature	*a*	*b*	R^2^	*a*	*b*	R^2^
50 °C	-	-	-	6.09	1.88 × 10^−2^	0.86
75 °C	79.7	6.56 × 10^−3^	0.79	4.81	5.24 × 10^−2^	0.99
50 °C-Pd	27.9	1.70 × 10^−1^	0.99	60.63	7.00 × 10^−2^	0.96

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
