# Peer review of "Improving Hazardous Gas Detection Behavior with Palladium Decorated SnO2 Nanobelts Networks"

_sensors, 2023, doi:10.3390/s23104783_

Round 1
Reviewer 1 Report
The authors present a detailed sensorial characteristic of the SnO2 nanowires-based gas sensors with and without palladium nanoparticle deposition. The manuscripts show that the Pd-decorated sensors has larger relative response, better response linearity, shorter response time, lower operating temperature.
I have several concerns.
1. How is the decorated Pd nanoparticle affect the sensor noise?
2. What is the Pd nanoparticle layer thickness used for Pd-decorated sensors characterization in Part 3.
3. From Figure 3, The Pd nanoparticle radius seems small and formed a relatively smooth layer different from previous publication which has large and well-separated Pd nanoparticles. How is the Pd particle size affect the sensor performance?
4. Why Pd-decorated SnO2 sensor can has much better response linearity than non-decorated SnO2 sensor? In Fig 7(c), the relative response in CO detection by the non-decorated SnO2 sensor at 50 degree even drops after approximately concentration of 300 ppm. Why such drop happens and how Pd decoration can prevent it?
Author Response
The answers are attached.

Reviewer 2 Report
The authors did a lot of good work, but ...
* what was really studied: nanowires or nanobelts, or maybe a mixture of both (paragraphs from line 207 and Fig. 2 b, line 472)? The reviewer knows from his experience that with thermal sublimation synthesis of metal oxides (such as SnO2) it is difficult to control the thermodynamics of the process and most often there is a mixture of different nanostructures: nanowires, nanobelts, nanorods, nanosails... In addition, as can be seen in Figure 3, the sizes of these structures are often not nanometer, but rather submicrometer.
* Fig. 1(b): very faintly visible,
* lines 176 and 177: what is nitrogen?
* caption Fig. 2 very confusing,
* no deciphering some of the acronyms of the names of the research equipment,
* lines 235 and 236: why suddenly such respect for oxygen, tin and palladium that capital letters were used?
* there is no figure 4c, which the authors refer to twice (by the way, in this paragraph from page 7 the calls are a bit defective: either bracket or "in Figure"),
* in the description of the test results presented in Figures 5 and 6, the reviewer lacks a reference to room temperature,
* line 391: the authors gave what they consider to be the response time, but did not specify when regeneration is achieved, i.e. return to the initial state,
Author Response
The answers are attached.

Round 2
Reviewer 1 Report
The authors have addressed my questions